# Automated Speech Analysis in Bipolar Disorder: The CALIBER Study Protocol and Preliminary Results

**DOI:** 10.3390/jcm13174997

**Published:** 2024-08-23

**Authors:** Gerard Anmella, Michele De Prisco, Jeremiah B. Joyce, Claudia Valenzuela-Pascual, Ariadna Mas-Musons, Vincenzo Oliva, Giovanna Fico, George Chatzisofroniou, Sanjeev Mishra, Majd Al-Soleiti, Filippo Corponi, Anna Giménez-Palomo, Laura Montejo, Meritxell González-Campos, Dina Popovic, Isabella Pacchiarotti, Marc Valentí, Myriam Cavero, Lluc Colomer, Iria Grande, Antoni Benabarre, Cristian-Daniel Llach, Joaquim Raduà, Melvin McInnis, Diego Hidalgo-Mazzei, Mark A. Frye, Andrea Murru, Eduard Vieta

**Affiliations:** 1Department of Psychiatry and Psychology, Institute of Neuroscience, Hospital Clinic of Barcelona, 08036 Barcelona, Catalonia, Spain; deprisco.michele@gmail.com (M.D.P.); cvalenzuela@recerca.clinic.cat (C.V.-P.); amasm@recerca.clinic.cat (A.M.-M.); violiva@recerca.clinic.cat (V.O.); giov.fico@gmail.com (G.F.); agimenezp@recerca.clinic.cat (A.G.-P.); lmontejo@recerca.clinic.cat (L.M.); megonzalezc@clinic.cat (M.G.-C.); popovic.dina@gmail.com (D.P.); pacchiar@clinic.cat (I.P.); valenti@clinic.cat (M.V.); mcavero@clinic.cat (M.C.); lcolomer@recerca.clinic.cat (L.C.); igrande@clinic.cat (I.G.); abenaba@clinic.cat (A.B.); dahidalg@clinic.cat (D.H.-M.); amurru@clinic.cat (A.M.); evieta@clinic.cat (E.V.); 2Bipolar and Depressive Disorders Unit, Digital Innovation Group, Institut d’Investigacions Biomèdiques August Pi i Sunyer (IDIBAPS), 08036 Barcelona, Catalonia, Spain; 3Biomedical Research Networking Centre Consortium on Mental Health (CIBERSAM), Instituto de Salud Carlos III, 28029 Madrid, Madrid, Spain; radua@recerca.clinic.cat; 4Department of Medicine, School of Medicine and Health Sciences, Institute of Neurosciences (UBNeuro), University of Barcelona (UB), 08007 Barcelona, Catalonia, Spain; 5Imaging of Mood- and Anxiety-Related Disorders (IMARD) Group, Institut d’Investigacions Biomèdiques August Pi i Sunyer (IDIBAPS), 08036 Barcelona, Catalonia, Spain; 6School of Graduate Medical Education, Mayo Clinic, Rochester, MN 55902, USA; joyce.jeremiah@mayo.edu (J.B.J.); majd.alsoleiti@mayo.edu (M.A.-S.); 7Office of Information Security, Mayo Clinic, Rochester, MN 55905, USA; chatzisofroniou.george@mayo.edu; 8Alix School of Medicine, Mayo Clinic, Rochester, MN 55905, USA; mishra.sanjeev@mayo.edu; 9School of Informatics, University of Edinburgh, Edinburgh EH16 4TJ, UK; filippo.corponi@ed.ac.uk; 10Mood Disorders Psychopharmacology Unit, University Health Network, Toronto, ON M5G 1M9, Canada; cristian-daniel.llachlopez@uhn.ca; 11Department of Psychiatry, University of Toronto, Toronto, ON M5S 1A8, Canada; 12Department of Psychiatry, University of Michigan, Ann Arbor, MI 48109, USA; mmcinnis@med.umich.edu; 13Department of Psychiatry and Psychology, Mayo Clinic, Rochester, MN 55905, USA; mfrye@mayo.edu

**Keywords:** bipolar disorder, speech analysis, natural language processing, predictive models, acoustic properties, language content, emotional profiles, diagnosis, precision psychiatry, global speech cohort

## Abstract

**Background**: Bipolar disorder (BD) involves significant mood and energy shifts reflected in speech patterns. Detecting these patterns is crucial for diagnosis and monitoring, currently assessed subjectively. Advances in natural language processing offer opportunities to objectively analyze them. **Aims**: To (i) correlate speech features with manic-depressive symptom severity in BD, (ii) develop predictive models for diagnostic and treatment outcomes, and (iii) determine the most relevant speech features and tasks for these analyses. **Methods**: This naturalistic, observational study involved longitudinal audio recordings of BD patients at euthymia, during acute manic/depressive phases, and after-response. Patients participated in clinical evaluations, cognitive tasks, standard text readings, and storytelling. After automatic diarization and transcription, speech features, including acoustics, content, formal aspects, and emotionality, will be extracted. Statistical analyses will (i) correlate speech features with clinical scales, (ii) use lasso logistic regression to develop predictive models, and (iii) identify relevant speech features. **Results**: Audio recordings from 76 patients (24 manic, 21 depressed, 31 euthymic) were collected. The mean age was 46.0 ± 14.4 years, with 63.2% female. The mean YMRS score for manic patients was 22.9 ± 7.1, reducing to 5.3 ± 5.3 post-response. Depressed patients had a mean HDRS-17 score of 17.1 ± 4.4, decreasing to 3.3 ± 2.8 post-response. Euthymic patients had mean YMRS and HDRS-17 scores of 0.97 ± 1.4 and 3.9 ± 2.9, respectively. Following data pre-processing, including noise reduction and feature extraction, comprehensive statistical analyses will be conducted to explore correlations and develop predictive models. **Conclusions**: Automated speech analysis in BD could provide objective markers for psychopathological alterations, improving diagnosis, monitoring, and response prediction. This technology could identify subtle alterations, signaling early signs of relapse. Establishing standardized protocols is crucial for creating a global speech cohort, fostering collaboration, and advancing BD understanding.

## 1. Introduction

Despite significant advances in bipolar disorder (BD) research, reliance on subjective clinical assessments for diagnosis [1,2] and monitoring persists [3]. People with bipolar disorder exhibit intense alterations in mood, energy, and thought [4], all of which are reflected in their speech patterns. Indeed, language, expressed through speech, serves as a privileged window into the mind; it is the foundation upon which we infer others’ thought processes and is thus the pillar of psychiatric evaluation. During clinical interviews, speech features are routinely assessed, albeit subjectively. These encompass acoustic features (e.g., tone, volume, prosody, and intonation), formal aspects (e.g., organization, flow, fluency, rhythm, quantity, and latency), as well as aspects of language content (e.g., coherence, cognitions, delusions, and obsessions), and finally emotionality (expressed feelings, affective tone) [5].

For instance, hypoprosody in depressed patients is usually identified by using acoustic features such as reduced variation in pitch (fundamental frequency), diminished changes in loudness (amplitude), and monotonous speech patterns [6,7]. Anxiety may be identified using the tremor of the voice, measured by jitter (variations in pitch) and shimmer (variations in loudness) [8], as well as by increased speech rate, irregular speech patterns, and higher vocal tension. Accelerated or decelerated thought rhythms in mania or depression are identified by assessing increased or decreased speech rates, respectively. Incoherent or circumstantial thought processes in psychosis or mania are identified by analyzing the semantics and syntax of speech. Depressed or elevated mood and guilt are identified by assessing the emotional tone of speech [9]. Indeed, we use speech features both for quantitative assessment of specific symptoms, as referenced in usual clinical scales, e.g., Young Mania Rating Scale (YMRS) [10] and Hamilton Depression Scale (HDRS) [11]), which allows us to establish syndromic and syndromal diagnoses.

Modern technology enables high-fidelity speech recording and subsequent analysis. Natural language processing (NLP) is a branch of artificial intelligence enabling machines to understand, interpret, and generate human language. In recent decades, the application of NLP techniques to analyze speech patterns in psychiatric disorders, including BD, has surged significantly [12].

Acoustic features of speech have demonstrated associations with most mental health diagnoses and many specific symptoms [8]. These include significant correlations between depressive [13,14] and manic symptoms in BD [15], as well as negative symptoms in schizophrenia [16]. They have proven effective in discriminating between depressed and non-depressed patients [17,18,19], as well as depression from bipolar disorder, schizophrenia, and healthy controls (HC) [20]. Similarly, they could accurately differentiate manic from depressed BD patients [7,21] and even showed the potential to predict depressive episodes [19].

Speech content analysis has emerged as a valuable tool for detecting subtle psychopathological changes that may elude the clinical ear, such as objectively quantifying speech incoherence, a hallmark of thought disorganization [22,23]. This method of analysis demonstrates a robust correlation with clinical scale scores and has proven effective in discriminating between stable schizophrenia patients and HC [22], as well as in predicting the transition to psychosis in high-risk populations [24,25]. Interestingly, speech content analysis has also shown utility in distinguishing manic BD patients from clinically stable schizophrenia patients [26] and in differentiating first-degree relatives of schizophrenia patients from HC [27]. Moreover, BD patients during hypomania showed increased verbal task switches and unique sound-based associations, distinguishing them from ADHD and HC [28].

Formal aspects of speech analysis in BD have allowed for discrimination between euthymia, mania, depression, and mixed episodes. By examining features such as speech organization, flow, fluency, rhythm, quantity, and latency, researchers can delineate distinct patterns associated with different phases of BD [29]. 

Emotional analysis is an NLP technique that aims to determine the emotional valence (positive, negative, or neutral) and content (such as joy, sadness, anger, fear, trust, disgust, surprise, and anticipation) conveyed in speech [30,31,32]. Studies have demonstrated that emotional analysis of speech contributes to predicting treatment response in resistant depression [33] and effectively discriminates between individuals with BD and HC [9]. These findings underscore the potential of emotional analysis to aid in the diagnosis, prognosis, and treatment monitoring of BD in the framework of precision psychiatry [34,35].

Speech features exhibit inter-individual variability, evident among HC and individuals with BD (as well as other mental health diagnoses), but notably, they also display significant intra-individual variability, particularly when comparing acute episodes of mania and depression with periods of euthymia. Analysis of speech features may reveal detectable alterations, offering a means for quantitative measurement and trait-state stratification within BD. To our knowledge, no studies have comprehensively explored speech in BD by combining analyses of these four features: acoustics, formal aspects, language content, and emotionality (e.g., sentiment and emotional tone). Some studies have moved in this direction, integrating some of these features, such as acoustics and emotionality [36], acoustics and semantic coherence [28], acoustics and formal aspects [37,38], or language content and emotionality [39]. In our current study, we aim to utilize these speech features not only for diagnosis and treatment outcomes but also to integrate them all and evaluate their relevance for each task.

We hypothesized that (i) speech features will correlate with the severity of manic and depressive symptoms, (ii) they will effectively differentiate between manic, depressive, and euthymic phases in BD, as well as between mania/depression and response, (iii) only specific speech features and speech tasks will be relevant for each of these analyses.

Our aims are (i) to correlate speech features with manic-depressive symptom severity in BD as measured by validated clinical scales, (ii) to use these speech features to develop predictive models for diagnostic purposes, capable of accurately distinguishing between manic, depressive, and euthymic phases in BD, and for predicting treatment outcomes by distinguishing between acute symptomatic phases and response, and (iii) to identify which specific speech features and speech tasks, or combinations thereof, are most relevant for each of these analyses.

## 2. Materials and Methods

### 2.1. Study Design

This is a naturalistic, observational study conducted in two centers from different countries (Spain and USA). Individuals diagnosed with BD experiencing manic and depressive episodes underwent audio recording during acute phases, and longitudinal recordings were also obtained after clinical response. Additionally, euthymic patients were recorded once. There were no disruptions to standard care or treatment as a result of the participation in the study, following the design of a study aimed at identifying digital biomarkers in BD [40,41,42]. Ethical approval was obtained in accordance with the ethical principles outlined in the Declaration of Helsinki [43] and Good Clinical Practice guidelines. The study protocol was reviewed and approved by the Ethics and Research Board of the recruiting centers (HCB/2020/0432 for Hospital Clínic of Barcelona and 22-010487 for Mayo Clinic Institutional Review Board) and complied with recommendations on studies on precision psychiatry [44]. Prior to their inclusion in the study, all participants provided written informed consent. Participation was entirely voluntary, and no incentives were offered to the patients.

### 2.2. Sample

#### 2.2.1. Hospital Clinic of Barcelona

A total of 65 patients diagnosed with BD were recruited in the Hospital Clinic of Barcelona (Barcelona, Catalonia, Spain). 

#### 2.2.2. Mayo Clinic

An additional 11 patients diagnosed with BD type I and in the acute manic phase were recruited from Generose Hospital at the Mayo Clinic (Rochester, MN, USA) as part of an independent study.

#### 2.2.3. Both Recruiting Centers

Aspects of the study design for the Hospital Clinic of Barcelona study are presented below with derivations in the Mayo Clinic study design noted in appropriate sections. Given the overall similarity of both projects combined with the scientific value of exploring linguistic analysis across multiple languages (i.e., Catalan, Spanish, and English) and patient populations, we present our collaborative effort between the teams.

The inclusion criteria comprised: (i) a diagnosis of BD (type I or II) confirmed through semi-structured diagnostic interviews [45], (ii) acute phases of (hypo)mania or depression as per DSM-5-TR criteria [46], or euthymia, defined by international consensus as sustained HDRS-17/YMRS scores ≤ 7 for at least 8 weeks [47]. The symptomatic response was defined as a ≥50% improvement in HDRS-17/YMRS scores, according to international consensus guidelines [47].

Exclusion criteria encompassed: (i) acute or organic dysphonia or other somatic comorbidities impacting speech (e.g., stroke, throat cancer), (ii) language impairment directly linked to treatment (such as lingual dystonia, tardive dyskinesia, sialorrhea), and (iii) psychiatric comorbidity (e.g., anxiety disorders, personality disorders, substance use disorders, ADHD) where these comorbidities resulted in symptom interference. This included psychiatric conditions that presented with symptoms severe enough to overshadow the primary affective (manic or depressive) symptoms of BD. This determination was made to ensure that the speech features analyzed in the study were primarily reflective of the affective states of BD rather than other psychiatric conditions. For example, severe anxiety might result in speech patterns characterized by nervousness or hesitation, which could confound the analysis aimed at distinguishing between manic and depressive episodes in BD patients. Notably, the presence of a psychiatric comorbidity was not an exclusion criterion per se when symptoms were not present or of minimal impact.

### 2.3. Assessment

#### 2.3.1. Sociodemographic and Clinical Assessment

##### Hospital Clinic of Barcelona

The Barcelona site collected the following sociodemographic and clinical variables: psychopathological status at inclusion (affective episode or euthymia) and date, patient factors (age, sex, type of BD, age of onset, first affective episode, number of previous affective episodes, number of psychiatric hospitalizations and reasons for admittance, suicide attempts); specifiers of the current episode (psychotic, anxious, mixed features, and suicidality); course specifiers (predominant polarity, rapid cycling, seasonal pattern); comorbidities (somatic and psychiatric); current and past drug use; treatment (psychopharmacological and other); and family history. 

##### Mayo Clinic

The Mayo Clinic study captured the date of the manic episode at inclusion and sociodemographic features felt to impact spoken language, including age, sex assigned at birth, gender, race, ethnicity, birth location, English fluency, highest level of education, occupational status, and household income.

#### 2.3.2. Symptoms and Functional Assessment

##### Hospital Clinic of Barcelona

Psychopathological symptoms were assessed using the following scales: manic symptoms with the YMRS [10], depressive symptoms with the HDRS-17 [11], positive and negative psychotic symptoms with the Positive and Negative Syndrome Scale (PANSS), where higher scores indicate more severe symptoms. Disease severity was assessed with the Clinical Global Impression-Severity (CGI-S) scale [48], where higher scores indicate greater disease severity. Functioning was evaluated with the Social and Occupational Functioning Assessment Scale (SOFAS) [49], which assesses functioning on a numeric scale from 1 to 100, with higher scores indicating better functioning, irrespective of symptom severity.

##### Mayo Clinic

Manic symptoms were assessed using the YMRS [10].

The rationale for using these specific scales in our study is based on their established validity, reliability, and widespread use in clinical and research settings for assessing various dimensions of psychopathology and functioning in BD. By using these validated instruments, we can ensure that our study results are reliable and comparable with other research in the field, thus enhancing the validity and generalizability of our findings.

### 2.4. Speech Recording

#### 2.4.1. Recording Method

##### Hospital Clinic of Barcelona

Interviews were recorded using a dual-channel lapel microphone system wirelessly transmitting to a receiver device connected to a laptop computer, acquiring signals at a frequency of 50 Hz to 20 KHz [50].

##### Mayo Clinic

Interviews were recorded using a head-worn miniature condenser microphone with a cardioid polar pattern (*C544L|AKG*) for the patient and a lapel-worn, lavalier microphone with a cardioid polar pattern (*Lv4-C|Movo Photo*) for the interviewer. Both microphones transmitted analog signals separately to an analog–digital converter (*Scarlett 2i2|Focusrite*), and the gain was calibrated to avoid clipping. Resultant digital files were saved in the wav format at a 48k sample rate and 24-bit depth.

#### 2.4.2. Language

Interviews were conducted in the patients’ native or preferred language, including Catalan, Spanish, or English. Although previous studies have not extensively analyzed speech across different languages, we do not anticipate significant issues for most parameters. This is because, first, intra-individual comparisons are made within the same language context, ensuring consistency. Second, many features of interest are language-independent, such as semantic coherence, prosody, and acoustic properties. For example, semantic coherence can be analyzed based on the logical flow and relevance of ideas, regardless of the specific language used. Therefore, despite the linguistic diversity, we expect the core speech features to be reliably analyzed across the different languages included in this study.

#### 2.4.3. Setting

Recordings were conducted in the typical clinical facilities of the hospital where patients receive treatment. These facilities include the inpatient psychiatric hospitalization unit for most patients experiencing manic episodes or severe depressive episodes and the outpatient mental health unit for most patients with hypomanic or mild-to-moderate depressive episodes, as well as all patients in euthymia. Recordings were performed using a standardized procedure to ensure consistency and reliability. This procedure included maintaining consistent microphone positioning, ensuring that lapel microphones were placed approximately 10–15 cm from the speaker’s mouth. The distance between the patient and the interviewer was kept at a standard 1–1.5 m, with both seated directly facing each other to facilitate clear communication. The evaluator’s position was also standardized, ensuring they were always seated in a manner that allowed for optimal audio capture without causing discomfort to the patient. Furthermore, room conditions, such as ambient noise levels and lighting, were kept consistent across all sessions. No acoustic isolation was used to prevent background noise or sound interferences, aiming to create a naturalistic setting that could be replicated in typical clinical care environments. Additionally, there was no physical separation between the interviewer and the patient, which may have resulted in some degree of overlapping audio during the recordings. This naturalistic approach was chosen to enhance the ecological validity of the study. The described setting conditions did not vary between the recruiting centers. Each recruiting center conducted recordings in the typical clinical facilities of the hospital where patients receive treatment, using the same standardized procedures to ensure consistency and reliability across all study environments.

#### 2.4.4. Interview Format

We conducted semi-structured interviews incorporating elements known to yield valuable insights into speech analysis in BD. The interviews comprised the following components, arranged in sequence (see Figure 1):

(i)Standard clinical evaluation—Participants were asked a variety of questions to complete the clinical scales for assessing psychopathological and functional states. Some of these scales include straightforward questions, such as item 16 from the HDRS, which asks about weight. Other items, like item 17 from the HDRS, require interpretation of responses to open-ended questions, similar to analyzing spontaneous speech. Clinical evaluations incorporating spontaneous speech have proven effective in detecting depression [51], identifying autism through acoustic feature analysis [52], and detecting manic states in BD [53];(ii)Cognitive task—Stroop test (approximately 3 min): Participants completed the Stroop test, which involves three main tasks. First, participants read aloud the names of colors printed in black ink. Second, they state the colors of the ink. Third, they perform the interference task, where they must state the color of the ink in which a color word is printed, ignoring the word itself (e.g., saying “red” when the word “blue” is written in red ink). This test assesses executive function–inhibition [54]. Mayo Clinic patients did not complete the Stroop test. The Stroop test has been used in previous literature studying prosodic features in depression [55], verbal task switches and unique sounds-based associations between BD, ADHD, and HC [28], and formal aspects of speech in BD discrimination between euthymia, mania, depression, and mixed episodes [29];(iii)Standard text reading (approximately 2 min): Patients were tasked with reading “The Rainbow Passage” [56], a 100-word excerpt commonly utilized by speech therapists to assess vocal ability. The Rainbow Passage has been used to evaluate acoustic markers as predictors of clinical depression scores [13] and fundamental frequency after a stressful activity [57];(iv)Non-emotional storytelling (approximately 3 min): Patients described the Cookie Theft picture, a visual scene depicted in a section of the Boston Diagnostic Aphasia Examination (BDAE)) [58]. This image was chosen to evoke a minimal emotional response. Patients were instructed to describe the image, including as much detail as they could, for at least one minute. If their response lacked sufficient content, supplementary questions were posed (e.g., “Please detail the steps for frying an egg, buttoning a button, putting on a shirt, or smoking a cigarette”). Non-emotional storytelling has been used to quantify speech incoherence in schizophrenia [23], detect incoherent speech in schizophrenia [22], and measure formal thought disorder in schizophrenia using image description [59];(v)Emotional storytelling (approximately 3 min): Patients were encouraged to recount autobiographical memories with emotional significance, such as discussing important childhood memories, significant individuals in their lives, moments of intense happiness or distress, future plans and expectations, and reflecting on how those memories have impacted them. Emotional storytelling has been used to distinguish between HC and patients with schizophrenia [60]. Furthermore, the emotional content of dreams has been shown to effectively differentiate between patients with BD, schizophrenia, and HC [26]. Notably, Mota et al. (2014) [26] demonstrated that speech containing emotional content is more valuable for discriminating between patients with BD, schizophrenia, and HC compared to speech without an emotional component.

All interviews adhered to a consistent structure, lasting approximately 40 min, and were conducted by mental health professionals (psychologists and psychiatrists). Interviewers employed clinical interview techniques such as paraphrasing and reflecting emotions to minimize their influence on the language content generated by participants, as described in previous studies [33,61].

### 2.5. Data Analysis

#### 2.5.1. Preprocessing

The recorded interviews underwent automatic diarization and transcription using a mixture of open-source [62,63] and proprietary software, with no data shared with third parties to ensure privacy. The diarization task automatically distinguished between the patient’s and the interviewer’s speech. Audio segments corresponding to the interviewer and any overlapping speech were removed to prevent interference in the analyses. Once the segments were identified, automatic transcription was performed. Each diarization and transcription step was followed by a manual review to check for errors and adjust software parameters for optimal performance, as the quality of interviews can vary, requiring parameter adaptations for accurate speaker identification. After transcription, personal information, such as names and family references, was automatically anonymized. The various parts of the interview (i–v) are identified using specific keywords (e.g., colors for the cognitive task), and this is verified manually. Each interview segment was then tagged for specific analyses.

#### 2.5.2. Feature Extraction

Acoustic features: Each interview was segmented into elements from conversational analysis, including turns, interpausal units, gaps, and pauses. For interpausal units, various acoustic features, including source, filter, spectral, and speech rate, were measured [8]. These features encompass measurements such as jitter, shimmer, harmonics-to-noise ratio, formant frequencies, Mel-frequency cepstral coefficients, and various aspects of pitch, intensity, and tempo. For gaps and pauses, various features based on the duration of silence were calculated, including delayed latency of responses, pause frequency, and pause length. The specific acoustic features are detailed in Table 1.

Language content (Syntactic-semantic features): Initially, pairs of questions and answers were segmented. Repetitions, fill-in words, and interjections (e.g., “ehm,” “aha,” etc.) and phrases made up entirely of stop words were removed. Phrases were tokenized to capture semantic meaning. Semantic coherence was extracted using methodology from previous literature populations [24,25]. Syntactic features such as syntactic complexity, sentence length, clause density, and the use of grammatical constructions were analyzed. Semantic features, including lexical diversity, referential clarity, thematic consistency, propositional density, use of abstract versus concrete language, and use of figurative language, were examined. Additionally, lexical–semantic relationships like synonymy, antonymy, hyponymy, hypernymy, collocations, and semantic fields were considered. The specific language content features are detailed in Table 2.

Formal aspects of language: Speech organization, flow, fluency, rhythm, quantity, and latency were extracted using previously described methods [29]. These features include coherence, cohesion, topicality, speech rate, articulation rate, disfluencies, smoothness, stress patterns, intonation, pacing, verbosity, word count, information density, response latency, onset time, and pause length. Additional aspects such as lexical richness, pronunciation accuracy, speech intelligibility, turn-taking, and the use of gestures and non-verbal cues were considered. The specific formal aspects’ features are detailed in Table 3.

Emotional features: The emotional content of the different parts of the interview was quantified, focusing on emotion words, sentiment analysis, intensity of emotion words [9,33], and the use of metaphors and figurative language. Prosodic features that convey meaning and emotion were also analyzed. The specific emotional features are detailed in Table 4.

The detailed methodologies for speech feature extraction will be provided in subsequent publications focused on each specific speech feature.

#### 2.5.3. Statistical Analysis

After data pre-processing and feature extraction, the following analyses will be conducted in accordance with the study objectives:Continuous Quantification of Psychopathology: Correlation of speech features with clinical scales assessing symptom severity for mania (YMRS), depression (HDRS-17), and psychosis (PANSS), including both global scores and specific items/symptoms;Categorical Classification: Using the speech features to develop predictive models for diagnostic (i.e., manic, depressive, and euthymic phases in BD) and treatment outcomes (i.e., acute phases of mania/depression vs. response phases). For these classification tasks, we will employ lasso logistic regression;Feature and Task Relevance Identification: The relevance of specific speech tasks and features (or combinations thereof) will be determined for each diagnostic and treatment outcome task. Variable relevance methods will be used to identify the most pertinent features. The magnitude of correlation and prediction accuracies across different speech tasks will be assessed to identify the most relevant tasks for the previous analyses.

#### 2.5.4. Code and Data Availability

The codebase was written in Python (version 3.11.9; Python Software Foundation), where the deep learning models were implemented in TensorFlow and developed on a single NVIDIA-GeForce RTX 4080 SUPER 16GB GDDR6X.

## 3. Results

A total of 76 patients diagnosed with BD have been enrolled in the CALIBER study. While the analysis of speech features is still ongoing, we will present here the main sociodemographic and clinical characteristics of the sample (see Table 5).

The average age of the participants was 46.0 ± 14.4 years, and the sample was predominantly female, with 48 women (63.2%). A notable 44 patients (67.7%) had non-psychiatric medical comorbidities, and 10 patients (15.4%) had psychiatric comorbidities. Past drug use was reported by 14 participants (21.5%), while 15 participants (23.1%) were current drug users.

The sample included 24 patients (31.6%) experiencing manic episodes, 21 patients (27.6%) in major depressive episodes, and 31 patients (40.8%) in a euthymic state. Among the patients in the acute phase, 15 out of 24 manic patients (62.5%) and 9 out of 21 depressed patients (42.9%) achieved a response. This distribution provided a balanced representation of the disorder’s different phases, allowing for comprehensive analysis across the spectrum of BD. 

Symptom severity varied significantly across the different phases. Manic patients had a mean YMRS score of 22.9 ± 7.1, indicating moderate to severe manic symptoms, reducing to 5.3 ± 5.3 (minimal to mild symptoms) after response. Depressed patients had a mean HDRS-17 score of 17.1 ± 4.4, reflecting moderate depressive symptoms, reducing to 3.3 ± 2.8 (minimal to mild symptoms) after response. The PANSS scores for these patients indicated mild to moderate psychotic symptom presence, with total symptoms averaging 50.4 ± 10.6. The mean CGI-S score was 4.3 ± 0.9, suggesting moderate to severe overall illness severity. The SOFAS score averaged 50.0 ± 13.0, highlighting the significant impact of acute episodes on functioning, typically suggesting moderate functional impairment.

In contrast, euthymic patients exhibited significantly lower symptom severity. Their mean YMRS score was 0.97 ± 1.4, and the HDRS-17 score was 3.9 ± 2.9, indicating minimal to mild symptoms. PANSS scores were also lower in this group, with total symptoms averaging 35.3 ± 5.2, reflecting minimal symptoms. The CGI-S score for euthymic patients was 1.7 ± 0.7, indicating mild illness severity. The SOFAS score was higher at 78.8 ± 9.9, reflecting good overall functioning during periods of euthymia.

Most recordings (52 patients, 80%) at the Barcelona site were conducted in an outpatient setting. Whereas the majority of recordings at the Mayo site (27 recordings, 87%) were conducted in the inpatient setting. Regarding treatment, a significant proportion of the patients were on psychopharmacological medications: 46 patients (70.8%) were receiving antipsychotics, 41 patients (63.1%) were on lithium, 29 patients (44.6%) were taking other mood stabilizers, 23 patients (38.5%) were on antidepressants, and 32 patients (49.2%) were using benzodiazepines. The variability of treatment between acute phases and response was low.

## 4. Discussion

The CALIBER study will represent a significant advancement at the intersection of psychiatric evaluation and modern technology in the context of BD. By leveraging the power of NLP and acoustic analysis, the study aims to enhance traditionally subjective clinical assessments with objective, quantifiable measures.

One of the most compelling aspects of the CALIBER study is its potential to improve diagnostic accuracy and treatment monitoring in BD. The use of speech features, such as acoustic properties, formal aspects, language content, and emotionality, provides a multi-dimensional view of a patient’s mental state. These features can objectively capture nuanced changes in speech patterns associated with different phases of BD, such as mania, depression, and euthymia. This objective measurement can complement traditional clinical evaluations, potentially leading to more precise and timely interventions [64,65], such as suicide prevention [66,67] and offering a tool to counterbalance therapeutic inertia in psychiatry [68].

Automated speech analysis offers a promising objective approach for accurately diagnosing mood episodes (manic, depressive, and euthymic) and predicting treatment outcomes. The longitudinal study of intra-individual changes in speech features will likely allow us to objectively measure subtle psychopathological changes that may be imperceptible to clinicians but indicate upcoming acute phases in BD. This knowledge may be used to train machine learning algorithms capable of predicting at-risk states, thus anticipating acute phases in BD and potentially allowing early intervention [69]. This is of utmost importance since acute episodes in BD often cause a high burden, functional limitations, and sometimes cognitive deficits. Prevention of mood episodes and early intervention are crucial to reducing their severity and duration, thereby mitigating the high impact of BD.

This study aims to determine which specific speech features, or combinations thereof, are most relevant for identifying specific symptoms (e.g., anxiety, irritability, thought disorganization, low mood) [70] and affective episodes (e.g., mania, depression, euthymia) in individuals with BD [20]. To achieve this, we will conduct a comprehensive analysis of various speech features, including acoustics, formal aspects, language content, and emotionality. This research is crucial because, while we currently understand specific associations between certain speech features and particular symptoms or episodes, we lack knowledge about which features, or combinations of features, are most relevant for each task. By identifying and prioritizing these speech features, we can potentially integrate them into automated algorithms for clinical use. Currently, it is not feasible to analyze all possible speech features simultaneously, which is why selecting the most relevant ones is essential.

Moreover, integrating speech analyses into clinical settings can complement routine consultations during the intervals between patient visits. These periods often involve significant uncertainty and bias due to the lack of information on the patient’s condition. By incorporating speech analyses into mobile phones, both patients and clinicians can continuously monitor symptoms between regular clinical interactions [71,72]. This real-time assessment of symptom fluctuations can be particularly valuable for early detection of relapses or responses to treatment, thereby enabling more proactive and personalized care [73].

One limitation of this study is the lack of previous research comprehensively exploring speech in BD by combining analyses of four key features: acoustics, formal aspects, language content, and emotionality. However, evidence from studies that have examined each modality individually, as well as improvements in patient identification and classification accuracy in studies combining various statistical and automated analysis methods [22,26], and those combining different analysis parameters within the same modality [19] support the feasibility of this approach. Additionally, the few studies focusing on multiple speech features simultaneously [28,36,37,38] further indicate that integrating these features can provide valuable insights. The comprehensive feature extraction from speech recordings, encompassing acoustic, syntactic, semantic, and emotional features, aims to provide a holistic analysis of speech, potentially leading to a more accurate and nuanced understanding and prediction of BD episodes.

The sample size in this study may appear relatively small compared to studies in other fields, such as genetics or neuroimaging. However, it is important to emphasize that this research focuses on speech digital data, where each patient contributes a substantial amount of information (e.g., interview recordings exceeding 30 min). This extensive data collection enables multiple analyses across various language features (see Table 1, Table 2, Table 3 and Table 4). Consequently, the study aligns with the principles of thick data studies, which involve an in-depth examination of a relatively small number of patients. The large volume of data permits detailed phenotypic characterization [74]. Additionally, this is a longitudinal study, meaning differences will be assessed using patients as their own controls. This design reduces the need for larger sample sizes. Supporting this approach, it is worth noting that most studies identifying significant differences in speech data within mental health populations typically include fewer than a few dozen participants [9,24,25,29,33].

A potential obstacle is the risk of overfitting in the machine learning models used for classification and feature relevance identification. While techniques such as cross-validation and feature selection are employed to mitigate this risk, overfitting remains a concern, especially given the relatively small sample size compared to the complexity of the data [75]. This can be addressed by implementing robust validation techniques, such as nested cross-validation, and by using regularization methods to prevent the models from becoming too complex. Additionally, we will perform extensive hyperparameter tuning and utilize ensemble methods to enhance model generalizability and reliability [76].

Another challenge is the inherent variability in speech that can be influenced by numerous factors unrelated to BD, such as environmental noise, physical health conditions affecting speech, individual differences in communication styles, and medication [77]. Although the study employs rigorous preprocessing and feature extraction techniques, these extraneous factors may still introduce variability that could confound the results. The inclusion of inter- and intra-individual comparisons allows for accounting of some of these factors, such as individual differences. Also, most patients included during acute manic phases are admitted to the inpatient unit, minimizing the variability of external conditions.

The exclusion criteria, while necessary to control for confounding variables, may also limit the generalizability of the findings. For instance, excluding individuals with psychiatric comorbidities or speech-affecting conditions could mean that the study’s findings are not fully representative of the broader BD population, many of whom have such comorbidities. However, this is needed to identify speech features associated with specific symptoms and affective phases. On the other hand, the naturalistic, observational design of the CALIBER study is a notable strength. By recording speech in typical clinical settings without altering standard care, the study ensures ecological validity. This approach enhances the generalizability of the findings to real-world clinical practice. Furthermore, the inclusion of a diverse sample from tow centers from different countries and using different largely spoken languages, such as Spanish and English, enhances the robustness and applicability of the findings across different populations and healthcare settings.

Moreover, there is significant variability in design among existing studies evaluating speech features in BD and other psychiatric disorders. This includes the interview format and the systems for data recording, processing, and analysis. This variability poses a challenge in establishing a standard design for the present study. Therefore, we have included a combined format for inter- and intra-individual comparisons. The longitudinal nature of the study allows for the assessment of intra-individual variability over time. This longitudinal approach is crucial for understanding how speech patterns change across different phases of BD and in response to treatment. Moreover, we have also included different types of interviews present in the literature, which have already yielded evidence of the association of speech features with specific symptoms or affective episodes [9,24,25,29,33]. 

To mitigate these challenges and promote consistency, it is essential to establish standardized protocols. Such protocols should encompass uniform interview formats, standardized data recording and processing systems, and consistent analytical methodologies. This standardization is critical not only for enhancing the reliability and validity of findings within individual studies but also for enabling meaningful comparisons and meta-analyses across different studies. By adhering to standardized protocols, researchers can build a global speech cohort, fostering collaboration and advancing our collective understanding of BD. Moreover, standardization facilitates the replication of studies and the validation of findings across diverse populations and settings, thereby enhancing the generalizability of results. This approach can lead to the development of robust, universally applicable diagnostic and monitoring tools for BD and other psychiatric disorders, ultimately improving patient outcomes on a global scale, following the lead of the Global Bipolar Cohort [78].

## 5. Conclusions

Automated speech analysis in BD might provide objective quantitative markers for psychopathological (manic/depressive) alterations. Using this technology we may be able to identify subtle alterations imperceptible to clinicians that represent early signs of relapse, allowing an early intervention. The implementation of this technology could potentially improve diagnosis, monitoring, and response prediction. Standardized protocols are crucial for establishing a global speech cohort, fostering collaboration, and advancing our understanding of BD.

## Figures and Tables

**Figure 1 jcm-13-04997-f001:**
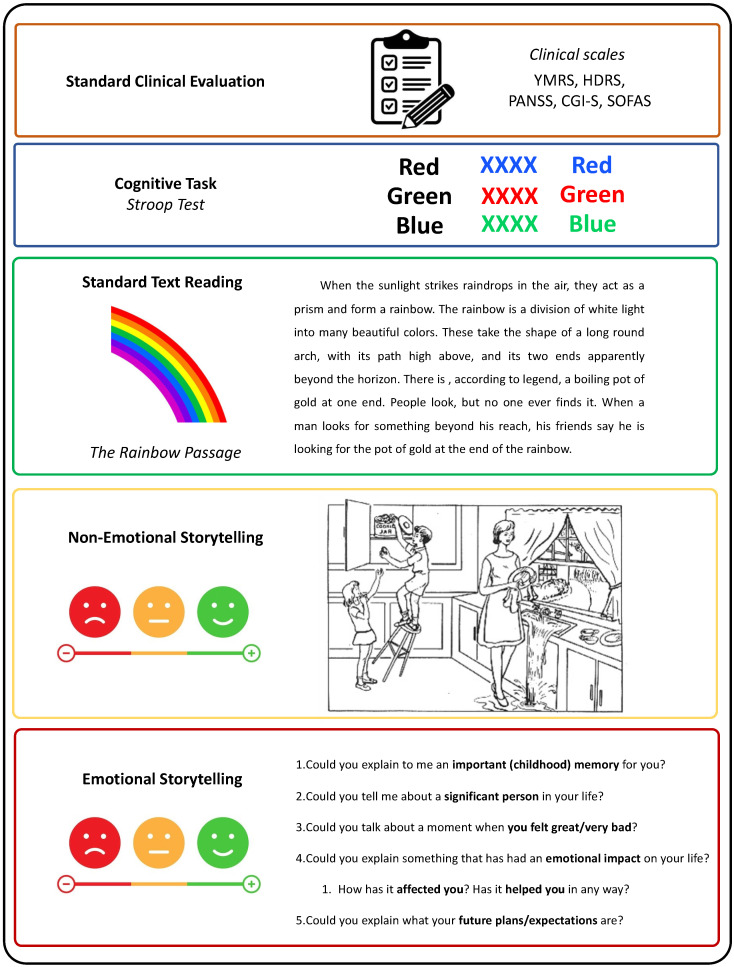
Structure of Semi-Structured Interviews. The semi-structured interviews comprised five key components to elicit diverse speech samples from participants: (i) Standard clinical evaluation—clinical scales; (ii) cognitive task—Stroop test: assessing executive function–inhibition (approximately 3 min); (iii) standard text reading: patients read “The Rainbow Passage” to evaluate vocal ability (approximately 2 min); (iv) non-emotional storytelling: describing the “Cookie Theft” picture from the Boston Diagnostic Aphasia Examination to evoke minimal emotional response (approximately 3 min); (v) emotional storytelling: recounting autobiographical memories with emotional significance (approximately 3 min). Interviews, lasting about 40 min, were conducted by mental health professionals using consistent techniques to minimize the interviewer's influence on participant language content.

**Table 1 jcm-13-04997-t001:** Acoustic Features.

Feature	Description
**Source Features**	
Jitter [%]	Deviations in individual consecutive f0 period lengths, indicating irregular closure and asymmetric vocal-fold vibrations.
Shimmer [%]	Difference in the peak amplitudes of consecutive f0 periods, indicating irregularities in voice intensity.
Tremor [Hz]	Frequency of the most intense low-frequency fundamental frequency-modulating component in a specified analysis range.
Harmonics-to-noise ratio (HNR) [dB]	Ratio between f0 and noise components, indirectly correlating with perceived aspiration.
Frequency disturbance ratio (FDR) [%]	Relative mean value of the frequency disturbance from 5 to 5 periods (five points average).
Amplitude Disturbance ratio (ADR) [%]	Relative mean amplitude value over a set of windows.
Quasi-open quotient (QOQ)	Ratio of the vocal folds’ opening time, often reduced in functional dysphonia.
Normalized amplitude quotient (NAQ)	Ratio between peak-to-peak pulse amplitude and the negative peak of the differentiated flow glottogram, normalized with respect to the period time.
Peak slope	Slope of the regression line that is fit to log10 of the maxima of each frame.
**Filter Features**	
F1 mean [Hz]	First peak in the spectrum of voiced utterances resulting from a resonance of the human vocal tract.
F2 mean [Hz]	Second peak in the spectrum of voiced utterances resulting from a resonance of the human vocal tract.
F1 variability [Hz]	Measures of dispersion of F1 (variance, standard deviation).
F2 variability [Hz]	Measures of dispersion of F2 (variance, standard deviation).
F1 range [Hz]	Difference between the lowest and highest F1 values.
Vowel space	F1 and F2 2D space for the vowels /a/, /i/, /u/.
Linear predictive coding (LPC) coefficients	Coefficients predicting the next time point of the audio signal using previous values.
**Spectral Features**	
Mel-frequency cepstral coefficients (MFCCs)	Coefficients derived by computing a spectrum of the log-magnitude Mel-spectrum of the audio segment.
**Prosodic Features**	
f0 mean [Hz]	Fundamental frequency, perceived as pitch (mean, median).
f0 variability [Hz]	Measures of dispersion of f0 (variance, standard deviation).
f0 range [Hz]	Difference between the lowest and highest f0 values.
Intensity [dB]	Acoustic intensity in decibels relative to a reference value.
Intensity variability [dB]	Measures of dispersion of intensity (variance, standard deviation).
Energy velocity	Mean-squared central difference across frames, possibly correlating with motor coordination.
Maximum phonation time [s]	Maximum time during which phonation of a vowel is sustained.
Speech rate	Number of speech units per second over the duration of the speech sample (including pauses).
Articulation rate	Number of speech units per second over the duration of the speech sample (excluding pauses).
Time talking [s]	Sum of the duration of all speech segments.
Utterance duration mean [s]	Mean duration of utterance length.
Pause duration mean [s]	Mean duration of pause length.
Pause variability [s]	Measures of dispersion of pause duration (variance, standard deviation).
Pause total [s]	Total duration of pauses.

**Table 2 jcm-13-04997-t002:** Language content (Syntactic–Semantic).

Feature	Description
**Syntactic Features**	
Syntactic Complexity	Degree of complexity in sentence structures, including the use of subordination and coordination.
Sentence Length	Average number of words per sentence.
Clause Density	Number of clauses per sentence.
Use of Grammatical Constructions	Frequency and variety of specific grammatical forms.
Part-of-Speech Distribution	Relative frequency of different parts of speech.
**Semantic Features**	
Semantic Coherence	Logical consistency and relevance of ideas within and across sentences.
Semantic Density	Amount of meaningful content per unit of speech.
Lexical Diversity	Variety of words used, measured by metrics such as type–token ratio.
Use of Abstract vs. Concrete Language	Proportion of abstract terms versus concrete terms.
Referential Clarity	Clarity with which entities are referred to and tracked throughout the discourse.
Thematic Consistency	Maintenance of a central theme or topic throughout a discourse.
Propositional Density	Number of propositions or ideas expressed per clause or sentence.
Use of Figurative Language	Frequency and types of non-literal language used.
Word Concreteness	Degree to which words refer to tangible, perceptible objects or experiences.
Sentiment and Emotion	Emotional tone conveyed through word choice.
**Lexical-Semantic Relationships**	
Synonymy	Use of different words with similar meanings.
Antonymy	Use of opposites to create contrast.
Hyponymy and Hypernymy	Use of specific terms and their general categories.
Collocations	Common pairings or groupings of words.
Semantic Fields	Grouping of related words that belong to the same domain of meaning.
**Discourse Features**	
Narrative Structure	Organization of content into a coherent story with elements such as setting, characters, plot, and resolution.
Argumentation and Reasoning	Use of logical arguments, evidence, and reasoning to support claims and ideas.
Topic Introduction and Maintenance	Ability to introduce new topics and maintain focus on them throughout the discourse.
Conclusion and Summarization	Effective wrapping up of discourse with a summary or conclusion.

**Table 3 jcm-13-04997-t003:** Formal aspects.

Feature	Description
**Speech Organization**	
Coherence	Logical arrangement of ideas in speech, ensuring it is easy to follow and understand.
Cohesion	Use of linguistic devices to link sentences and parts of discourse together.
Topicality	Relevance of the content to the topic at hand, maintaining focus without unnecessary digressions.
**Flow and Fluency**	
Speech Rate	Number of speech units per second, including pauses. Note: Also listed under Acoustic Features.
Articulation Rate	Number of speech units per second, excluding pauses. Note: Also listed under Acoustic Features.
Disfluencies	Interruptions in the flow of speech, such as filled pauses, repetitions, and self-corrections.
Smoothness	Degree to which speech is uninterrupted and flows naturally.
**Rhythm**	
Stress Patterns	Distribution of emphasis on syllables within words and across phrases.
Intonation	Variation in pitch across an utterance.
Pacing	Timing and spacing of speech sounds and silences.
**Quantity**	
Verbose vs. Concise	Amount of speech produced relative to what is necessary.
Word Count	Total number of words spoken in a given time frame or speech segment.
Information Density	Amount of information conveyed per unit of speech.
**Latency**	
Response Latency	Time taken to respond to a question or prompt.
Onset Time	Time from the beginning of an utterance to the start of the first spoken word.
Pause Length	Duration of pauses within speech.
**Additional Speech Features**	
Lexical Richness	Variety and sophistication of vocabulary used.
Pronunciation Accuracy	Correctness of phoneme production.
Speech Intelligibility	Clarity of speech, making it understandable to listeners.
Turn-Taking	Ability to appropriately manage and transition between speaker and listener roles.

**Table 4 jcm-13-04997-t004:** Emotional Features.

Feature	Description
Emotion Words	Use of specific words that convey emotions (e.g., joy, sadness, anger, fear, surprise, trust, etc.).
Sentiment Analysis	Overall positive or negative sentiment of the speech content.
Intensity of Emotion Words	Degree of emotional intensity conveyed through word choice (e.g., “furious” vs. “angry”).
Metaphors and Figurative Language	Use of metaphors or similes to convey emotions (e.g., “I feel like I’m walking on air” to express happiness).
Prosodic Features	Variations in pitch, loudness, and duration that convey meaning and emotion. Note: Also listed under Acoustic Features.

**Table 5 jcm-13-04997-t005:** Characteristics of the sample.

	Hospital Clinic of Barcelona		Mayo Clinic	
	Acute Phase	Response	Acute Phase	Response
**Total patients recruited**				
Manic Episode (acute phase) N (%)	13 (20)	9 (69.2)	11 (100)	8 (73)
Major Depressive Episode (acute phase) N (%)	21 (32.3)	9 (42.9)		
Euthymia N (%)	31 (47.7)			
Total N (%)	65 (100)		11 (100)	
**Symptoms and functional variables**				
**Patients with acute episodes**				
YMRS score (manic patients only) (M ± SD)	24 ± 8.5	5.9 ± 6.2	21.7 ± 5	4.6 ± 4.3
HDRS-17 score (depressed patients only) (M ± SD)	17.1 ± 4.4	3.3 ± 2.8		
PANSS positive symptoms score (M ± SD)	11.0 ± 7.3	8.5 ± 3.0		
PANSS negative symptoms score (M ± SD)	12.1 ± 5.1	9.7 ± 4.5		
PANSS general symptoms score (M ± SD)	27.3 ± 5.5	21.2 ± 4.2		
PANSS total symptoms score (M ± SD)	50.4 ± 10.6	39.4 ± 7.8		
CGI-S score (M ± SD)	4.3 ± 0.9	2.4 ± 1.3		
SOFAS score (M ± SD)	50.0 ± 13.0	69.1 ± 23.9		
**Euthymic patients**				
YMRS score (M ± SD)	0.97 ± 1.4			
HDRS-17 score (M ± SD)	3.9 ± 2.9			
PANSS positive symptoms score (M ± SD)	7.0 ± 0.2			
PANSS negative symptoms score (M ± SD)	8.7 ± 3.1			
PANSS general symptoms score (M ± SD)	19.6 ± 3.2			
PANSS total symptoms score (M ± SD)	35.3 ± 5.2			
CGI-S score (M ± SD)	1.7 ± 0.7			
SOFAS score (M ± SD)	78.8 ± 9.9			
**Sociodemographic and clinical variables**				
Age (M ± SD)	48.1 ± 13.3		33.6 ± 14.5	
Sex: Females N (%)	42 (64.6)		6 (54.5)	
Age of Onset (M ± SD)	32.9 ± 10.9		26.8 ± 12.2	
Illness Duration (years) (M ± SD)	14.9 ± 12.6		7.8 ± 7.3	
Number of Previous Affective Episodes (Median, IQR)	1, 1–2		4, 2–6	
Psychotic Features (patients on acute episodes only) N (%)	9 (24.3)		0 (0.0)	
Anxious Features (patients on acute episodes only) N (%)	29 (78.4)		11 (100.0)	
Mixed Features (patients on acute episodes only) N (%)	9 (24.3)		6 (54.5)	
Active Suicidality (patients on acute episodes only) N (%)	12 (32.4)		3 (27.3)	
Non-Psychiatric Medical Comorbidities N (%)	44 (67.7)		10 (90.9)	
Psychiatric Comorbidities N (%)	10 (15.4)		9 (81.8)	
Past Drug Use N (%)	14 (21.5)		6 (54.5)	
Current Drug Use N (%)	15 (23.1)		7 (63.6)	
**Setting**				
Outpatient N (%)	52 (80)		4 (13)	
**Psychopharmacological Treatment**				
Antipsychotics N (%)	46 (70.8)		11 (100)	
Lithium N (%)	41 (63.1)		7 (63.6)	
Other Mood Stabilizers N (%)	29 (44.6)		5 (45.4)	
Antidepressants N (%)	23 (38.5)		1 (9.0)	
Benzodiazepines N (%)	32 (49.2)		8 (0.72)	

## Data Availability

The original contributions presented in the study are included in the article; further inquiries can be directed to the corresponding author.

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
