# Peer review of "Automated Speech Analysis in Bipolar Disorder: The CALIBER Study Protocol and Preliminary Results"

_jcm, 2024, doi:10.3390/jcm13174997_

Round 1

Reviewer 1 Report

Comments and Suggestions for Authors

The article, entitled Automated Speech Analysis in Bipolar Disorder: The CALI-2 BER Study, addresses the use of natural language processing analysis technologies in mental health diagnosis, an area of relevant interest.

On review, it became evident that the article was more akin to a research protocol than a research report.

The primary issue is the absence of results from the language processing analysis using technology, which renders the article unsuitable for consideration.

If you wish to publish a research protocol, it should be prepared in accordance with the relevant standards. However, such publications are not within the scope of JCM's objectives. In this case, the document does not meet the criteria for either a protocol or an empirical article that reports results in line with the stated in his title, abstract or objectives.

Furthermore, the following questions were raised in the order in which they appear:

The title and abstract do not adequately reflect the content of the research protocol.

The number of authors and their respective contributions do not appear to be commensurate with the content of the manuscript.

The sources are not entirely reliable. Many of the sources are from technology conferences, but this is not reflected in the text. It should be noted in the text that the sources are from technology conferences. On the other hand, the sources used in relation to language processing are somewhat outdated and/or classical, and do not fully align with the current technological landscape. Overall, the document appears to attempt to bolster its arguments with a sense of confidence by using augmentative language, despite a lack of sufficient evidence.

In terms of methodology, the study is described as multicentre, however, it is in fact a bicentre study, with two centres and one partially participating. It would be more logical and precise to report a study in Spain and USA.

While the procedure is well established, the precise methods for extracting language features are not clearly defined. While examples are provided, they do not include the specific methods that will be employed. Accordingly, there is no information available regarding the reliability of these methods or the applications or outcomes they have previously demonstrated.

With regard to the results, the authors have only provided data on the clinical and socio-demographic characteristics. No results from the analysis of language processing have been included.

It is important to note that only partial results are reported for the Mayo Clinic group (mania and sociodemographics), which means that the study would then be primarily based on the responses of 65 participants.

It is evident that the discussion section does not address the outcomes of the study, which renders the issues raised therein irrelevant. Furthermore, the authors do not address the arguments pertaining to the differences in languages, which in this case, are three. They also fail to address the number of participants and their external validity, as well as the language processing methodologies. All of them are basic elemments of discussion.

It is, however, anticipated that the research will yield results.

Reviewer 2 Report

Comments and Suggestions for Authors

Dear authors,

Thank you for this report outlining your novel approach to identifying speech differences in Bipolar Affective Disorder. The findings seem likely to be promising to improve the precision in detecting and differentiating affective episodes. I congratulate the authors for this commendable effort.

I have made comments on the manuscript PDF which I believe could improve the report overall. However, I note one major point:

This report introduces itself as a speech analysis study and extensively features methods utilised for this type of analyses. However, the actual results reported do not include any of those analyses, as I understand they are still being finalised. In this case, I would either recommend authors hold back from publishing any findings from this until they have the full set of results or amend the structure of the manuscript to clearly define the report as an introduction of the CALIBER study and its sample characteristics. As it stands, the discussion is of little relevance as it focuses on the speech analysis - which is basically not reported in the manuscript.

I hope I made my point clear

Thanks again for the opportunity to review this paper:

line 40-44: This report is about a limited aspect of the CALIBER study, however this is no clear in your title and aims. Please amend these appropriately to reflect the actual scope of the current report.

Materials and Methods: I recommend summarising the site-specific differences either under one separate section or for each materials-methods section.

line 162: Was there any incentivisation involved?

line 168-170: Given the sample size, this seems to me a limitation rather than a strength of the design.

line 183: How was the symptom interference determined? What is the exhaustive list of psychiatric comorbidities that were excluded?

line 187: nature or date of this episode? or both?

line 191: please see above comment on psychiatric comorbidity.

line 196: Please briefly explain the rationale of using these particular scales.

line 224-227: Please expand on how all features were objectively measured and why language spoken may or may not lead to significant differences in these components.

line 230: Please explain in further detail the differences and similarities across study centres. As it stands, it is not clear whether both environments were sufficiently similar/standardised.

line 249: Please explain why these specific tasks were used and how they relate to which of the speech components intended to analyse.

Round 2

Reviewer 1 Report

Comments and Suggestions for Authors

The authors have addressed all aspects of the report. The point is: it is still a research protocol rather than an empirical article, but now at least it is clearly marked.